Controlled-release of Bacillus thurigiensis formulations encapsulated in light-resistant colloidosomal microcapsules for the management of lepidopteran pests of Brassica crops

Bashir Oumar 1
http://orcid.org/0000-0001-7363-1186 Claverie Jerome P. 1 jerome.claverie@usherbrooke.ca
Lemoyne Pierre 2
Vincent Charles 2 Charles.Vincent@agr.gc.ca
1 Université de Sherbrooke , Sherbrooke, Qc , Canada
2 Horticultural Research and Development Center, Agriculture and Agri-Food Canada , Saint-Jean-sur-Richelieu, Qc , Canada
Huber Dezene
Electronic publication date: 2016 Oct 11
Publication date: 2016
Volume: 4
Electronic Location ID: e2524
Received 2016 Apr 23; Accepted 2016 Sep 3
Copyright: © 2016 Bashir et al.
Copyright year: 2016
Copyright holder: Bashir et al.
License: This is an open access article distributed under the terms of the Creative Commons Attribution License, which permits unrestricted use, distribution, reproduction and adaptation in any medium and for any purpose provided that it is properly attributed. For attribution, the original author(s), title, publication source (PeerJ) and either DOI or URL of the article must be cited.
License URL: https://creativecommons.org/licenses/by/4.0/

Keywords: Bacillus thurigiensis, Microcapsules, Encapsulation, Biopesticide, Brassica crop

Funding: Agriculture and Agri-Food Canada A-base funding program of Agriculture and Agri-Food Canada financially supported this work. The funders had no role in study design, data collection and analysis, decision to publish, or preparation of the manuscript.

==============================
Bacillus thuringiensis (B. t.) based formulations have been widely used to control lepidopteran pests in agriculture and forestry. One of their weaknesses is their short residual activity when sprayed in the field. Using Pickering emulsions, mixtures of spores and crystals from three B. t. serovars were successfully encapsulated in colloïdosomal microparticles (50 μm) using innocuous chemicals (acrylic particles, sunflower oil, iron oxide nanoparticles, ethanol and water). A pH trigger mechanism was incorporated within the particles so that B. t. release occurred only at pH > 8.5 which corresponds to the midgut pH of the target pests. Laboratory assays performed on Trichoplusia ni (T. ni) larvae demonstrated that the microencapsulation process did not impair B. t. bioactivity. The best formulations were field-tested on three key lepidopteran pests that attack Brassica crops, i.e., the imported cabbageworm, the cabbage looper and the diamondback moth. After 12 days, the mean number of larvae was significantly lower in microencapsulated formulations than in a commercial B. t. formulation, and the effect of microencapsulated formulations was comparable to a chemical pesticide (lambda-cyhalothrin). Therefore, colloïdosomal microcapsule formulations successfully extend the bioactivity of B. t. for the management of lepidopteran pests of Brassica crops.

Introduction

Bacillus thuringiensis (B. t.) is an aerobic bacterium which upon sporulation forms a parasporal inclusion body, the crystal. The latter is made of Cry proteins which often exhibit insecticidal activities (Höfte & Whiteley, 1989). Worldwide, B. t. based formulations account for ca. 50% of the market for sprayable biopesticides (Research Markets, 2013). One weakness of such formulations is their short residual activity in the field, resulting from UV light-induced degradation of the toxin (Zhou, She & Liu, 2015). This shortcoming can be addressed by developing innovative formulations which screen or reflect UV light.

The formulation of a pesticide is of paramount importance to optimize its efficacy. Ideally, the formulation should provide maximal effect to the active ingredient, while exerting minimal unintended negative effects on non-target organisms. Knowles (2008) outlined the main factors to take into account in the formulation of agrochemicals. Briefly, these factors are the physico-chemical properties of the active principle, its biological activity and mode of action, the method of application, the safety in use, the formulation cost, and the market preferences. Microencapsulation has recently attracted considerable attention because of the possibility of controlling the release rate of an active component. Microencapsulated formulations are now used for a variety of applications, such as pharmaceutical (Wise, 2000), biotechnological (Ma, 2014; Murua et al., 2008) and agricultural ones (Kim et al., 2012; Linder & Markus, 2005). However, most of the methods used to fabricate microcapsules for biopesticides require environmentally unfriendly conditions, such as the use of organic solvents and monomers or elevated temperatures. As mentioned by Boyetchko et al. (1999) and Brar et al. (2006), the microencapsulation of biopesticides based on living organisms (e.g. bacteria, fungi, viruses) is often accompanied by a loss of bioactivity. In their review on nanoparticle systems for the delivery of pesticides, De et al. (2014) only mentioned inert active ingredients (i.e., chemicals or proteins). Behle et al. (2011) developed a formulation whereby a hydrophobic fungus (Beauvaria bassiana) and a hydrophilic baculovirus (Pieris rapae granulovirus) were encapsulated in a UV-protecting microcapsule. Likewise, when sprayed on apple trees in an orchard, DiPel (a commercial B. t. serovar kurstaki formulation) and a bio-encapsulated formulation caused significant mortality to oblique banded leafroller (Choristoneura rosaceana Harris) larvae respectively up to 3 and 14 days after treatments (Côté et al., 2011).

Here we present a novel B. t. encapsulation system based on the fabrication of Pickering emulsions. Pickering emulsions are stabilized by interfacial nanoparticles. For example, in a water-in-oil (w/o) Pickering emulsion, micron-size droplets of water are dispersed in a continuous oil phase and nanoparticles are located at the interface between the water and oil phases. Depending on their surface tension, these nanoparticles can stabilize the emulsion in the colloidal sense. They prevent the fusion of water droplets which eventually leads to phase separation. Velev, Furusawa & Nagayama (1996a) and Velev, Furusawa & Nagayama (1996b) were the first to demonstrate that Pickering emulsions can be transformed into microcapsules when the interfacial nanoparticles are either linked or fused together to form a continuous wall (Fig. 1). This step, usually referred as locking, can be achieved via a variety of techniques, such as chemical cross-linking (Thompson et al., 2010), in situ polymerization (Bon & Colver, 2007), thermal softening (Dinsmore et al., 2002) and solvent-instability of the particle (Keen, Slater & Routh, 2012a; Keen, Slater & Routh, 2012b; Nomura & Routh, 2010). When nanoparticles are polymeric in nature, a continuous wall can thus be formed around the aqueous droplet, leading to the formation of a microcapsule called colloïdosome by analogy with liposomes which are phospholipid stabilized aqueous vesicles.

Figure 1 (A) Schematic representation of the preparation process for the B. t. strains encapsulated within the colloïdosomal microcapsules. (1) latex particles of poly(butylmethacrylate-co-methacrylic acid) in water. (2) latex + B. t. spores and crystals (3) sunflower oil + Fe3O4 nanoparticles. (4) W/O Pickering emulsion. (5) Locking of the microcapsules upon addition of ethanol. (6) Collection of the microcapsules by centrifugation. In the actual process, ethanol is added at the same time as the oil phase, and therefore, the Pickering emulsion (4) is not isolated. (B) Representative pictures of the various steps of the Pickering process. Step 6a corresponds to the microcapsules immediately after centrifugation. The microcapsules are at the bottom of the vial, and are surmounted by an aqueous solution and then by the oil solution. Once the oil is separated, the redispersed microcapsules appear as a free flowing powder in water (6b).

In this work, we prepared novel pH-sensitive colloïdosomal microparticles containing B. t. parasporal crystals (i.e., mixtures of spores and crystals) using a solvent-instability locking mechanism recently developed by Nomura & Routh (2010). These microparticles were designed so that 1) they maintained the bioactivity of B. t.; 2) they could release B. t. parasporal crystals in the midgut of a lepidopteran larvae, notably when pH > 8; 3) they were opaque in order to prevent light-induced damage of the B. t. spore; and 4) they could be dispersed in water in order to form a free-flowing non-viscous aqueous formulation.

Four key lepidopteran pests attack Brassica crops (i.e., broccoli, Brussel sprouts, cabbage and cauliflower) in Canada: the imported cabbageworm (Artogeia rapae (L.)), the cabbage looper (Trichoplusia ni, T. ni (Hübner)), the diamondback moth (Plutella xylostella (L.)) and the alfalfa looper (Autographa californica Spreyer) (Agriculture Agri-Food Canada, 2012). Here, we document the development of a microencapsulated formulation of three bioactive B. t. serovars. The efficiency of these formulations was first assessed through a series of laboratory assays on cabbage looper larvae, and, in a second step, these formulations were tested in a cabbage field against aforementioned Brassica pests.

Materials and Methods

B. t. culture and preparation

The bacterial strains B. t. serovar kurstaki HD1, B. t. serovar aizawai, B. t. serovar tolworthi and B. t. serovar 407 (Sheppard et al., 2013) were grown in CCY medium (Stewart et al., 1981) and incubated in a rotary shaker (200 rpm) at 30 °C for 3 days. All B.t. cultures were centrifuged at 8,000 rpm for 15 min at 4 °C. The pellets, composed essentially of spores and parasporal inclusion bodies mixture, were washed with sterilized bidistilled water (ddH20) and precipitated with 5% lactose weight/weight and acetone to obtain a dry and fine powder soluble in water (Dulmage, Correa & Martinez, 1970). After microscopic examination, more than 90% of the cells were lyzed.

Preparation of latex particles of poly(butylmethacrylate-co-methacrylic acid)

In a 250 mL glass-jacketed reactor was added 100 mL of nanopure water (σ = 18 MΩ/cm), 10 g of butyl methacrylate (70 mmoL, purified over basic alumina), 5 g of methacrylic acid (59 mmoL, purified over silica), 0.5 g of sodium dodecyl sulfate as stabilizer and 0.25 g of potassium persulfate as initiator. This mixture was mechanically stirred at 400 rpm and degassed by bubbling nitrogen for 20 min. It was then heated to 80 °C while keeping a nitrogen blanket above the liquid phase in order to avoid O2 introduction. After 12 h of reaction time, a latex devoid of floc was obtained. The particles were readily soluble in alkaline environment (pH = 8.5). The latex solid content, as determined gravimetrically, was 12%.

Formation of the colloïdosome

The preparation process is outlined in Fig. 1 and representative pictures are presented in Fig. 2. In a first incubation tube was added 20 mL of commercial sunflower oil, 10 mg of Fe3O4 nanoparticles (size = 40 nm, see Supporting Data, measured by dynamic light scattering (DLS) in water) and 2 mL of absolute ethanol (Fig. 1A, Step 3). The mixture was processed with a vortex mixer for 1 min. In a separate tube, 1 g of B. t. kurstaki strain (under the form of a dry powder) was added to 5 mL of the latex prepared in 2.2 (Fig. 1A, Step 2). Then, 5 mL of nanopure water containing 146 mg of NaCl was added and the mixture was stirred with a vortex mixer for a minute. The contents of both vials were added to each other and were mixed again with a vortex mixer for one minute (Fig. 1B, Step 6a). Then, the mixture was centrifuged at 5,000 rpm (Fig. 1B, Step 6b) in order to separate the dark grey colloidosomal microcapsules from the supernatant oil phase (Fig. 1A, Step 6). In order not to break the microcapsules, centrifugation was never performed for more than 5 min, but the process was repeated until oil was no longer observed on top of the microcapsules.

Figure 2 (A) Latex of PolyA as observed by AFM.

(B) Strain of B. t., as observed by AFM. (C) Colloidosomal microcapsules containing the encapsulated B. t., as observed by optical microscopy (20X). (D) Kinetics of dissolution of a microcapsule at pH = 8.5, as observed by optical microscopy (10X). The green bar (50 µm) is set at the same location in each picture.

Characterization by atomic force microscopy (AFM)

The suspension containing the microcapsules was diluted 100 times with water. A drop of the diluted suspension was applied on a mica plate and was left to dry in air. The plate was analyzed on a Veeco Dimension 5000 microscope equipped with Nanoscope V controller (Bruker/Veeco, Santa Barbara, CA, USA). All images were taken at room temperature using tapping mode and analysed with the Gwyddion software.

Insect rearing, laboratory assays, field trial and persistence of formulations

Rearing

A cabbage looper colony was maintained at Agriculture and Agri-Food Canada at 25 °C, 70% R.H., and 16 h photophase/8 h scotophase. Larvae were fed on solidified diet prepared according to Shorey & Hale (1965).

Laboratory assays

Liquid diet was poured in stainless steel molds (18 × 9 × 9 cm). Upon solidification of the diet, 1 cm thick slices were cut and put on a glass plate. A plastic grid was enforced in the diet, thus enclosing 50 cells (1 × 1 cm) with diet in the bottom half. From a suspension containing B. t. (microencapsulated or not) at a concentration of 1 mg/mL, a 50 μl aliquot was pipetted on the surface of the diet of each cell. It was allowed to dry for 1 h. One first instar cabbage looper larva was put on the treated diet of each cell. The preparations were held at 25 °C, 70% R.H., and 16 h photophase/8 h scotophase for 7 days. The negative controls consisted of water, iron nanoparticles, B. t. serovar 407 (a Cry- B. t. serovar) and lactose powder. The positive control was B. t. kurstaki HD1. Larval mortality was assessed after 7 days.

Persistence of B. t. formulations

Cabbage seedlings (8–9 leaves) grown in pots in greenhouses were brought to a field at the Experimental Farm of Agriculture and Agri-Food Canada in L’Acadie (45,29972 N, 73,34972 W), Quebec, Canada. They were randomly assigned to one of six treatments and sprayed accordingly with a handheld sprayer until run-off on 29 July 2015 (= day 0). The treatments were: 1) no treatment (negative control 1); 2) microencapsulated B. t. serovar 407 (cry-) (negative control 2); 3) microencapsulated B. t. serovar kurstaki HD-1; 4) microencapsulated B. t. serovar aizawai; 5) microencapsulated B. t. serovar tolworthi; 6) Bioprotec CAF (commercial Bt-based biopesticide, 8.2% dry weight) (AEF Global, Lévis, Canada) (positive control 1); The formulations were applied at a concentration of 1 g of B. t. (not formulated)/L. Bioprotec CAF was applied at a rate of 2.8 L/ha. Thus, 72 potted plants were assigned as follows: 6 treatments × 6 durations in the field × 2 plants per combination treatment/duration. Immediately after the sprays, two plants were selected for each treatment and were brought back to the laboratory for assays (= day 0). Once the sprays dried, the pots containing the remaining plants were buried in the soil, leaving the plants exposed to normal climatic conditions.

At day 1, 2, 3, 6 and 9 after application, ten leaves were picked from two plants randomly selected in each treatment and four discs were punched out of each leaf. These discs were deposited in 9 cm Petri dishes, and 10 neonate T. ni from a mass rearing were introduced in the dish for a total of 2 × 50 larvae per treatment. Larvae were allowed to feed on the discs for 24 h, and then transferred onto untreated artificial diet (Shorey & Hale, 1965). Larvae were maintained at 25 ± 1 °C, 60–70% R.H. and L:D 16:8 (Sanyo model MLR-351H; Sanyo, Osaka, Japan). Larval mortality was assessed five days after transfer to artificial diet.

Field trial

The field trial was done a few meters away from the persistence trials. Cabbage seedlings (cv. “Lennox”) were transplanted at 3–4 true leaf stage in a Saint-Blaise clay loam on 30 June 2014. They were spaced 47 cm apart in the rows and the rows were spaced 80 cm apart. The experimental layout was a complete randomized block with four replicates. Each block had 28 rows and covered 257.6 m2 (11.5 × 22.4 m). It was subdivided into seven experimental units consisting of four contiguous rows and each corresponded to a treatment randomly assigned within a given block. Blocks were spaced by a non-cultivated 5 m wide strip.

The six treatments described above were conducted and a seventh chemical pesticide treatment (positive control two) was conducted using the insecticide Matador® (Lambda-cyhalothrin; Syngenta Canada Inc., Guelph, Canada), applied at 83 mL/ha. The treatments were applied on 28 August and 5 September 2014 and only the two central rows of each experimental unit were sprayed in order to leave a buffer between treated rows.

The B. t. serovars were prepared as described above. The powders were formulated as described in Section 2.2. The surfactant Tween 80® was added to the tank mix at a concentration of 0.1% v/v. The treatments were applied at a rate of 500 L/ha with a SOLO backpack sprayer (model 425), at a maximal pressure of 200 kPa (2 bars) (SOLO, Newport News, VA, USA). The mean flow was 1.28 L/min.

One day before insecticidal sprays, i.e., 27 August 2014, larval infestations of the imported cabbageworm (P. rapae), the diamondback moth (P. xylostella), and the cabbage looper (T. ni), were determined. Weekly counts of larvae on seven randomly selected plants in each experimental unit were made to evaluate the larval abundance and to determine the efficacy of treatments until 9 September 2014.

Temperature, precipitation and solar radiation were taken from a weather station on the experimental farm located ca. 600 m from the efficacy trial. Relevant data was extracted form a database with the software CIPRA (Bourgeois et al., 2008).

Insect counts of the field trial and arcsin-transformed mortality data of the persistence assays were analyzed using an ANOVA treatment followed by a Tukey HSD test for post-hoc comparisons of means. All statistical analyses were done with XLSTAT Version 2011.5.01 (Addinsoft, New York, NY, USA).

Results and Discussion

Preparation of the colloidosomes

The preparation of the colloidosomal microcapsules began with the preparation of a pH-sensitive latex of poly(butylmethacrylate-co-methacrylic acid) (polyA) by an emulsion polymerization process (Fig. 1A, Step 1). The latex was constituted of polymer particles with a hydrodynamic diameter of 143 nm as measured by DLS (see Supporting Data), and in good agreement with the average diameter of 183 nm measured on 100 individual dry particles by AFM (Fig. 2A). Several reasons motivated our choice for polyA, which is the main constituent of the walls of the colloidosomal microcapsules. First, acrylic polymers are currently accepted by the U.S. Food and Drug Administration for use in contact with food, according to the code of federal regulations title 21 (Code of Federal Regulations, 2015). Second, the emulsion polymerization process is performed in water in the absence of any organic solvent. Third, at the alkaline pH of the midgut of T. ni (pH ∼8.5) (Wang & Kelly, 1985) the polymer is dissolved in water, and the content of the capsule is released. However, at lower pH such as those generally encountered in the environment, the polymer is insoluble in water, and the envelope of colloïdosomal microcapsule is preserved. Indeed, the capsules are intact at slightly acidic or neutral pH (Fig. 2C), but dissolution of the polymeric capsule wall at pH = 8.5 is observed in less than 15 s by optical microscopy, as shown in Fig. 2D. Last, the glass transition temperature (Tg) of polyA, as measured by differential scanning calorimetry (see Supporting Data), is only 65 °C. The Tg value corresponds to the temperature at which a polymer switches from hard and glassy to soft and viscous. This low value favors the locking process (fusion of the latex polymer particles in a continuous polymer shell). Once the latex was prepared, B. t. bacterium and NaCl were added. The B. t. bacterium appeared in the form of an ellipsoid of 2 μm length and 1 μm diameter, as measured by AFM (Fig. 2B). The role of NaCl was to stabilize the emulsion against Ostwald ripening in the subsequent step. The chosen oil phase was alimentary sunflower oil. Black iron oxide (Fe3O4) nanoparticles (1 g/L) were added in the oil phase to serve as light absorbing material in the final formulation. Iron oxide nanoparticles have been reported to be devoid of toxicity at low doses (Samanta et al., 2008). They exhibit a strong absorption in the 300–900 nm region, with a broad maximum at 440 nm, as measured by UV-visible absorption spectroscopy. Upon mixing the aqueous phase (B) and the oil phase (C), a Pickering w/o emulsion was obtained (D). The latex nanoparticles have a surface energy which is intermediate between oil and water, thus they position themselves at the interface between water and oil. Upon adding ethanol, the latex particles within the aqueous droplet become colloidally unstable. They coalesce to form a continuous shell of polymer, resulting in the formation of a microcapsule. The dark grey capsules were separated from the oil phase via gentle centrifugation. Separation from the oil phase was found to be necessary, as oil was found to be toxic to the T. ni larvae, thus inducing false positive results (see below). The capsules had an average diameter of 50 microns (Figs. 2C and 2D). Notably, the gap between the mandibles (measurement of inner structures) of first instar T. ni larvae was measured to ca. 110 microns, thus large enough to allow the passage of the microparticles. Once separated from the supernatant, the microparticles appeared as a free-flowing grey powder. Remarkably, the fabrication of these capsules did not require any organic solvent and was entirely performed at room temperature, which was necessary to maintain B. t. bioactivity. As negative control, microcapsules containing lactose powder instead of B. t. were also prepared in the same fashion.

Laboratory bioassays

In laboratory bioassays, larval mortality of T. ni was < 10% for Control 1 (water) and 2 (iron nanoparticles). Acetonic powder of B. t. 407 caused < 14% larval mortality (Table 1) which is expected because this B. t. serovar has no bioactivity on T. ni (Sheppard et al., 2013). Lactose powder caused 5.9–9.6% larval mortality when formulated without oil, while it was 72–76% when formulated with oil. Similarly, mortality caused by microencapsulated acetonic powder of B. t. 407 was much higher when formulated with oil (> 75.5%) than in the absence of oil (< 28.6%). These results demonstrate that except for oil, all other components present in the colloïdosome are innocuous to T. ni larvae. All further results are reported for colloïdosomes which are free of oil. By contrast, when the microcapsule contained B. t. kurstaki HD-1 powder, ca. 100% larval mortality was observed, which was comparable to the mortality observed with non-encapsulated B. t. kurstaki HD-1 powder. Our results demonstrate that the bioactivity of B. t. kurstaki HD-1 was not significantly altered by the microencapsulation process.

Table 1 Mortality of T. ni larvae following treatment with various B. t. formulations in the laboratory.

	Bioassay 1	Bioassay 2	
Treatment	# larvae assayed	% larval mortality	# larvae assayed	% larval mortality	
Control 1 (water)	51	0.0	53	7.5	
Control 2 (iron nanoparticules)	50	2.0	51	9.8	
B. t. 407	50	0.0	50	14.0	
B. t. 407 (microencapsulated, no oil)	50	0.0	49	28.6	
B. t. 407 (microencapsulated, with oil)	53	75.5	51	92.2	
Lactose	50	4.0	50	14.0	
Lactose (microencapsulated, no oil)	51	5.9	52	9.6	
Lactose (microencapsulated, with oil)	50	76.0	50	72.0	
B. t. kurstaki HD-1	50	100.0	50	100	
B. t. kurstaki HD-1 (microencapsulated, no oil)	49	75.5	48	97.9	
B. t. kurstaki HD-1 (microencapsulated, with oil)	49	100.0	50	100	

Table 2 Meteorological conditions prevailing during (A) persistence and (B) field tests with B. t. formulations at L’Acadie (Qc, Canada) in 2014.

(A) Persistence test, treatment of July 29, 2014 (= day 0)	
Days after treatment	Maximum temperature (°C)	Minimum temperature (°C)	Average temperature (°C)	Relative humidity (%)	Precipitations (mm)	Solar radiation (MJ/m2)	
0	20.4	11.6	15.9	81.4	0.0	23.9	
1	23.7	10.7	17.9	79.8	0.0	25.2	
2	22.0	13.1	17.4	88.2	8.8	18.2	
3	25.8	13.6	20.1	81.6	0.0	21.3	
4	27.3	15.5	20.6	88.3	0.0	17.3	
5	27.6	14.1	21.1	80.2	0.0	24.3	
6	27.4	16.3	21.5	79.9	0.0	20.7	
7	25.5	14.7	20.2	84.7	0.0	19.4	
8	25.1	15.0	19.8	68.2	2.2	24.4	
9	23.3	14.3	18.0	82.5	0.0	18.3	
(B) Field test, treatment of August 28, 2014 (= day 0)	
Days after treatment	Maximum temperature (°C)	Minimum temperature (°C)	Average temperature (°C)	Relative humidity (%)	Precipitations (mm)	Solar radiation (MJ/m2)	
0	20.7	11.8	17.3	74.7	0.0	13.2	
1	21.7	9.8	16.1	78.1	0.0	21.6	
2	25.0	12.4	20.3	77.3	0.0	18.1	
3	23.4	18.8	21.2	92.7	12.6	4.9	
4	26.4	17.2	21.5	87.2	0.0	16.7	
5	28.7	17.6	22.5	84.9	1.2	14.0	
6	24.6	16.1	19.9	79.0	1.0	20.0	
7	27.2	13.0	20.0	80.6	0.0	19.4	
8	28.6	17.4	23.5	81.0	2.0	15.8	
9	24.3	14.2	20.2	85.4	4.4	5.1	
10	21.6	8.2	14.9	78.5	1.1	18.6	
11	24.0	8.6	16.8	76.8	0.0	18.2	
12	23.0	11.8	17.5	69.6	0.0	19.6	

Field trials and persistence of the colloidosomal B. t. formulation

In the persistence trial, % larval mortality was low for the control (water) and for B .t. 407 (microencapsulated), averaging 10.3% (with an outlier of 37% on the third day after treatment) (Fig. 3). Immediately after treatment, the highest larval mortality (97.9%) was observed for B. t. kurstaki (microencapsulated). With larval mortality of 95.8, 88.7, 70.9, 38.4 and 23.2% determined respectively after 1, 2, 3, 6 and 9 days, the bioactivity of that treatment remained consistently higher than any others. The second best persistence was observed for B. t. tolworthi (microencapsulated), with 77.0, 79.9, 41.4, 40.8, 28.3, and 14.3% larval mortality at 0+, 1, 2, 3, 6 and 9 days after treatment respectively. The Bioprotec CAF treatment (non-encapsulated B. t. kurstaki) caused 32.9 and 31.4% larval mortality immediately and 1 day after treatment, respectively, while it caused < 10% larval mortality from day 3 to day 9. Thus, the bioactivity of B. t. is significantly extended by microencapsulation.

Figure 3 Persistence of bioactivity of B. t. formulations sprayed on potted cabbage plants in the field on 29 July 2014 (= day 0).

Leaf disks were punctured and brought in the laboratory where they were assayed on T. ni larvae (n = 50 per treatment, two repeat experiments, error bar: standard deviation).

One day before treatment (i.e., 27 August 2014) in the field trial, all larval populations of lepidopteran pests consistently averaged ca. 8 larvae per plant (Fig. 4). Throughout the field test, larval populations for the control (water) and for B. t. 407 (microencapsulated) were not significantly different and ranged from 7.2 to 9 larvae per plant. Six days after treatments (i.e., 3 September 2014), larval populations in the commercial formulation (Bioprotec CAF) were not significantly different from the control (water) and B. t. 407. Larval populations for the microencapsulated B. t. kurstaki, B. t. aizawai and B. t. tolworthi were lower than for control experiments, ranging from 2.8–4.2 larvae per plant, while lambda-cyhalothrin was significantly lower than all other treatments (1.5 larvae per plant). Twelve days after treatments (i.e., 9 September 2014), there were no significant differences between lambda-cyhalothrin, microencapsulated B. t. kurstaki, B. t. aizawai and B. t. tolworthi (all having ca. 1.4 larvae per plant), while there was significant a difference between the latter treatment and Bioprotec CAF (ca. 4 larvae per plant). Thus, the microencapsulated B. t. formulations are as efficient as the broad range pyrethroid insecticide Matador, and significantly more efficient than non-encapsulated B. t. formulations.

Figure 4 Average number of lepidopteran (all species and larval stages pooled), larvae per cabbage plant in the field.

A first evaluation of larval populations was done one day before treatment, i.e., 27 August 2014. Treatments were done on 28 August and 5 September 2014 (See Table 2 for meteorological conditions). For a given number of days after treatment, numbers flanked by the different letters are significantly different at p < 0.05 (Tukey HSD).

Conclusions

In this paper, we demonstrated that the Pickering emulsion process is an efficient process for the encapsulation of biopesticides using only innocuous components. Thus, opaque B. t.-loaded colloidosomes were shown to maintain the bioactivity of B. t., even when exposed to solar illumination. These colloidosomes included a pH-triggered release mechanism in order to deliver the payload in the slightly alkaline environment of lepidopteran midguts. In theory, this pH-triggering mechanism would preclude unintended impacts upon non-target organisms having rather acidic digestive systems. Field-tests demonstrated that the microencapsulation process confer extended B. t. bioactivity. Based on these positive results, we believe that the use of colloïdosomal microencapsulated formulations is a promising strategy for the development of environmentally acceptable pesticides for agriculture systems.

We thank Guy Boivin (Agriculture and Agri-Food Canada, Saint-Jean-sur-Richelieu, QC, Canada) for T. ni specimens and Michel Brouillard, Jérémie Côté, Corentin Moreau and Vicky Lachance for technical input. We thank Jean-François Landry, Ottawa Research and Development Centre (Agriculture and Agri-Food Canada) for help in measuring structures of the head of T. ni larvae.

Additional Information and Declarations

Competing Interests

Author Contributions

Data Deposition

The authors declare that they have no competing interests.

Oumar Bashir performed the experiments, analyzed the data, contributed reagents/materials/analysis tools. This contribution was performed at the Université du Québec à Montréal.

Jerome P. Claverie conceived and designed the experiments, analyzed the data, wrote the paper, prepared figures and/or tables, reviewed drafts of the paper. Part of this contribution was performed at the University of Québec at Montréal.

Pierre Lemoyne performed the experiments, analyzed the data, contributed reagents/materials/analysis tools, prepared figures and/or tables.

Charles Vincent conceived and designed the experiments, analyzed the data, wrote the paper, prepared figures and/or tables, reviewed drafts of the paper.

The following information was supplied regarding data availability:

Zenodo: http://doi.org/10.5281/zenodo.154446.

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
