# Peer review of "Controlled-release of Bacillus thurigiensis formulations encapsulated in light-resistant colloidosomal microcapsules for the management of lepidopteran pests of Brassica crops"

_PeerJ, doi:10.7717/peerj.2524_

## Round 0.1 · original submission · Minor Revisions

· Academic Editor

Minor Revisions

Both reviewers were positive about the MS, which describes the use of microencapsulation to increase efficacy of a biopesticide. The authors use basic statistical methods which are appropriate for the study. Reviewer 1 mentions reporting variability, and I agree. Both reviewers provided useful comments that the authors must carefully consider in their revision and rebuttal. Reviewer 1 also provided an edited MS that corrects some mainly minor language issues.

Thank you to the reviewers for their careful assessments, and to the authors for submitting a good and useful study.

Reviewer 1 ·

Basic reporting

I have suggested a few changes in the text to reflect standard English grammar. However, the article was quite well-written overall, and the background, materials and methods, results, and conclusions were very clear.

Experimental design

The study was very well designed and reflected the authors' understanding of proper controls (they were impressive in this respect!) as well as true replication. I would like to have seen a higher level of true replication but understand the limits of space and other resources in this type of investigation. The research questions were well defined and they address a knowledge gap that was fully described and has a solid basis in the current state of the science, at least as far as I know it. The study has shown the efficacy of using colloidosomal microcapsules made using Pickering emulsions for encapsulating B.t. serovars in a formulation that screens UV light and exploits the insect's alkaline gut pH to trigger rupture of the microcapsules and release of the toxin.

Validity of the findings

The data are robust and were given proper controls (both positive and negative). Significant differences were apparent at relevant points, and the analyses were appropriate for this type of data. My only concern, indicated in the reviewed draft that is attached, is that measurements of variation for the results were not reported in the text nor in the graphics. This lack should be remedied before publication.

Additional comments

I'm personally very glad to see this type of work. Many benign active ingredients fail to be used because of formulation limitations, and this work removes a significant hurdle for deployment of many biopesticides.

Annotated reviews are not available for download in order to protect the identity of reviewers who chose to remain anonymous.

Reviewer 2 ·

Basic reporting

This manuscript (MS) authored by Bashir et al presents results on Bt formulations encapsulated in UV-resistant microcapsules and its usefulness to control lepidopteran agricultural pests. The MS includes both results of laboratory tests and field tests results. Controlled laboratory tests show that microencapsulation not reduce the insecticidal activity of Bt for T. ni larvae while field trials show that the insecticidal activity of Bt is significantly extended by microencapsulation. It is a systematic and comprehensive study. This is a well presented and written manuscript with only a few very minor comments. It will contribute significantly to this field, given that one of the weaknesses of Bt formulations is their short residual activity when sprayed on field.

Experimental design

No comments

Validity of the findings

The authors found that "the capsules had an average diameter of 50 microns" (see line 245). They performed laboratory tests using Trichoplusia ni neonate larvae to show that the microencapsulation does not deteriorate Bt insecticidal activity. To validate these results, the MS should include a paragraph where it is discussed in detail that the size of the mouth of the neonate T. ni larvae, or other lepidopteran pests, is large enough for ingestion of particles of 50 microns.

Additional comments

Line 15 – spores must be replaced by parasporal crystals or mixture of spores and crystals which is the active ingredient of the Bt formulations.
Line 27 – Brassica should be written in italics
Line 109 – What is the serovar 407? The serovar is named with a word that usually refers to where the bacteria was isolated (eg. Israelensis, mexicanensis, ...), the name of a scientific personality (aizawai, kurstaki, ..), etc.
Lines 156 and 163 – Information on Bt serovar 407 is confusing since in some parts of the text is said to be "a non-sporulent Bt serovar" (see line 156), while in others it is said to be a Bt strain not poducing parasporal crystal (see line 163).

---

## Round 0.2 · accepted · Accept

· Academic Editor

Accept

Thank you for your revisions and rebuttal. The initial reviewers' comments all pointed to minor revisions, and the authors have done a good job at responding/rebutting those minor concerns. This paper is now ready for publication in PeerJ.